# The Role of Hydrogen Peroxide and Peroxiredoxins throughout the Cell Cycle

**DOI:** 10.3390/antiox9040280

**Published:** 2020-03-26

**Authors:** Sukyeong Heo, Suree Kim, Dongmin Kang

**Affiliations:** Department of Life Science, Fluorescence Core Imaging Center, and Research Center for Cellular Homeostasis, Ewha Womans University, Seoul 03760, Korea; hcjm123@gmail.com (S.H.); kimsuree@hanmail.net (S.K.)

**Keywords:** hydrogen peroxide, peroxiredoxin, cell cycle, cyclin-dependent kinase, centrosome

## Abstract

Hydrogen peroxide (H_2_O_2_) is an oxidizing agent that induces cellular damage at inappropriate concentrations and gives rise to an arrest during cell cycle progression, causing cell death. Recent evidence indicates that H_2_O_2_ also acts as a promoter for cell cycle progression by oxidizing specific thiol proteins. The intracellular concentration of H_2_O_2_ is regulated tightly, enabling its use as a cellular signaling molecule while minimizing its potential to cause cellular damage. Peroxiredoxins (Prxs) have peroxidase activity toward H_2_O_2_, organic hydroperoxides, and peroxynitrite for protecting cells from oxidative stress. They are suggested to work as signaling mediators, allowing the local accumulation of H_2_O_2_ by inactivating their peroxidase activity uniquely compared with other antioxidant proteins such as catalase and glutathione peroxidase. Given that Prxs are highly sensitive to oxidation by H_2_O_2_, they act as sensors and transducers of H_2_O_2_ signaling via transferring their oxidation state to effector proteins. The concentrations of intracellular H_2_O_2_ increase as the cell cycle progresses from G_1_ to mitosis. Here, we summarize the roles of Prxs with regard to the regulation of cell cycle-dependent kinase activity and anaphase-promoting complex/cyclosome in terms of changes in H_2_O_2_ levels. Protection of the cell from unwanted progression of the cell cycle is suggested to be a role of Prx. We discuss the possible roles of Prxs to control H_2_O_2_ levels.

## 1. Introduction

The role of H_2_O_2_ as a signaling molecule that regulates various biological processes, such as cell proliferation, differentiation, and migration, is well established [1,2]. Relationships between the change in H_2_O_2_ concentrations and cell cycle progression are less well understood because of limited information for specific effectors of H_2_O_2_ at different stages throughout the cell cycle. Constitutively elevated levels of H_2_O_2_ caused by Nox1 overexpression can transform cells and can make them tumorigenic [3], indicating that H_2_O_2_ is a promoter of cell proliferation. Persistent production of H_2_O_2_ by treatment with extracellular reactive nitrogen species via possible activation of epidermal growth factor receptors [4] and subsequently activated nicotinamide adenine dinucleotide phosphate (NADPH) oxidases [5,6] or with glucose oxidase in media containing high concentrations of glucose induces cell cycle arrest at the boundary between the G_0_ and G_1_ phase [7,8]. Reactive oxygen species (ROS) are required for G_1_–S phase transition during cell cycle progression [9]. A study [10] in human melanoma cell lines reported the downregulation of ROS levels with treatment of diphenyleneiodonium or vitamin E and reported that Nox4 lockdown using siRNA oligos induces cell cycle arrest at the G_2_–M stage. However, the molecular mechanism by which ROS controls transition of specific stages during cell cycle progression remains to be elucidated.

Peroxiredoxins (Prxs) are abundant intracellular H_2_O_2_-removing enzymes that have the conserved peroxidatic cysteine (C_P_) as the site of oxidation by H_2_O_2_, forming cysteine sulfenic acid (C_P_-SOH) and resulting in the formation of a disulfide bond with the resolving cysteine (C_R_) [11,12]. The disulfide bond (C_P_-S-S-C_R_) is subsequently reduced by thioredoxin or glutaredoxin [13,14]. In mammalian cells, Prxs constitute six Prx isoforms (PrxI–PrxVI) that localize in various places, including the nucleus, cytosol, mitochondria, endoplasmic reticulum (ER), peroxisome, and extracellular space. Considering the fluctuation of levels of ROS throughout the cell cycle [15,16], whether the accumulation of H_2_O_2_ by the regulation of Prx activity occurs and what compounds serve as effectors of H_2_O_2_ are subjects of importance that need to be addressed.

The pericentrosomal-associated PrxI is inactivated by cell cycle-dependent kinase 1 (Cdk1) at the G_2_–M transition, which allows an increase in H_2_O_2_ levels at the centrosome [17]. This is critical for a positive feedback activation of Cdk1 during mitotic progression. The E3 ubiquitin ligase anaphase-promoting complex/cyclosome (APC/C) plays important roles in cell cycle progression by degrading key cell cycle regulators, such as cyclin A, cyclin B, geminin, and securin, appropriately in a cell cycle stage-dependent manner [18,19,20]. Activity of APC/C is suggested to be controlled by H_2_O_2_ through oxidizing and inactivating APC11, a subunit of APC/C [21], or by oxidizing Cdc14B, a positive regulator of APC/C–Cdh1 [17]. Activities of both Cdks and APC/C, two key cell cycle effectors, are regulated by H_2_O_2_. Given that H_2_O_2_ induces oxidative damage to cells, the cell produces detoxifying enzymes such as Prxs and glutathione peroxidase. During cell cycle progression, the coordinated balance between the organelle-associated H_2_O_2_-producing system and the H_2_O_2_-removing Prxs is expected to be important for the regulation of cell cycle effectors including Cdks.

In this review, we describe the roles of Prxs associated with intracellular organelles with regard to controlling cell cycle progression and regulation of Cdks and APC/C by H_2_O_2_. We discuss the association of Prx1 with the centrosome for proper mitotic entry and local regulation of Prx1 by Cdk1 during mitosis. We consider the possibility that H_2_O_2_ molecules from the fragmentation of the Golgi, ER, and mitochondria are used as important regulators for cell cycle progression by inactivating Prxs locally.

## 2. Dynamic Change in Intracellular Morphology and Oscillation of Activities of Cdks and APC/C through the Cell Cycle

Regulation of the cell cycle has been attributed largely to oscillation of Cdk activities. In eukaryotes, about 20 different Cdk family proteins and four major cyclins (cyclin D, cyclin E, cyclin A, and cyclin B) exist. Cdk activities are regulated by the phosphorylation state of Cdks; by the concentrations of Cdk inhibitors, such as the Ink4 family (p16, p15, p18, and p19) and the Cip/Kip family (p21, p27, and p57) [22,23]; and by the amount of cyclins, which is increased in a transcription-dependent manner and is decreased by ubiquitin–proteasome pathway through the APC/C and the skp–cullin–F-box complex [24]. APC/C degrades two important cyclins (cyclin A, a cofactor for Cdk2 and Cdk1; cyclin B, a cofactor of Cdk1) at early mitosis and at late mitosis, respectively, to ensure an irreversible and directional cell cycle progression. Broadly, APC/C activity is inversely proportional to the activities of Cdk1 and Cdk2 throughout the cell cycle, except at the G_2_ phase (Figure 1) [18,19]. At the G_2_ stage, early mitotic inhibitor 1 inhibits APC/C through its binding with Cdc20 and Cdh1, two activators of APC/C [25,26]. Detection with 2’,7’-dichlorodihydrofluorescein, which fluoresces on oxidation, showed that levels of ROS increase at G_1_–S phase transition, peak at mitosis, and decrease during mitotic exit to complete the redox cycle [9,16]. However, this probe shows several limitations, such as relatively low selectivity to react with various oxidants including peroxynitrite, nitric oxide, and hypochloride in addition to H_2_O_2_, and susceptibility to photooxidation and photobleaching [27,28,29]. At late G_1_–S phase transition, APC/C–Cdh1 activity is inhibited by increased endogenous ROS, but a direct target of ROS has not been investigated [9]. APC11, a subunit of APC/C, is sensitive to oxidation by H_2_O_2_ [21] but is not known to be an effector of endogenous H_2_O_2_.

The structure of intracellular organelles is changed dramatically at G_2_–M phase transition (Figure 1). The mammalian Golgi apparatus is structured in stacks of flattened membrane cisternae and plays a central role in protein and vesicular trafficking from the ER to the plasma membrane, the endosomes, or secretion outside of the cell [30]. Golgi fragmentation is required for mitotic entry [31] and induced through phosphorylation by polo-like kinase 1 (Plk1) [32], Mek1 [33], and subsequently Cdk1 [34]. In cells activated with growth factors, Golgi resident Duox1 and Duox2 proteins are responsible for production of H_2_O_2_ at the Golgi [35]. Activation of Duox1 and Duox2 proteins requires elevation of intracellular Ca^2+^ concentration. Considering that 1) a transient increase in Ca^2+^ concentration at mitotic spindle poles is observed [36,37,38] and 2) fragmented Golgi containing Duox1 and Duox2 are absorbed into the ER, a major reserve organelle for Ca^2+^, H_2_O_2_ molecules are produced by Ca^2+^-dependent activated Duoxs and can be used as signaling mediators during mitosis. Mitochondria are implicated in a variety of events, including energy metabolism through the production of ATP and the supply of metabolites, apoptosis by the release of cytochrome C and anti-inhibitor of apoptosis (anti-IAP), calcium homeostasis, and the production of ROS [39]. The morphology of mitochondria is changed dynamically throughout the cell cycle by fusion and fission. Mitochondria fuse to form a hyperactive long tubular network at the G_1_ phase and become the fragmented (granular) short-length state that facilitates their segregation at mitotic stage [40,41]. Mitochondria are a major cellular source of H_2_O_2_, which results from incomplete reduction of O_2_ during the steps of the electron-transfer chain [42,43]. Increased ROS from damaged mitochondria enforce a G_1_–S cell cycle checkpoint as signaling mediators by activating mitogen-activated protein kinase signaling and thereby increasing an inhibitor of Cdk2–cyclin E in *Drosophila melanogaster* [44]. A hyperfused mitochondrial state is linked to the G_1_–S transition with proper accumulation of cyclin E [40], indicating that a signal from mitochondria controls cell cycle progression. The dynamin-related protein 1, a promoter of mitochondrial fission, is activated by cyclin B–Cdk1 complexes, thereby leading to mitochondrial fragmentation at the G_2_–M transition [45]. The dynamin-related protein 1 is degraded by APC/C–Cdh1 during mitotic exit [46]. The Cdks and APC/C modules regulate structural dynamics of mitochondria, Golgi, and ER. Retrograde signals from the organelles can regulate the activities of Cdks and APC/C. H_2_O_2_ molecules are suggested to be signaling mediators in the mutual interaction between cell cycle controllers and change in organelle structure.

## 3. Localization of Peroxiredoxin Proteins Inside and Outside of the Cell

Peroxiredoxins are a major class of antioxidant enzymes that reduce H_2_O_2_ with the use of electrons derived from NADPH [47]. Mammalian cells express six Prx isoforms (PrxI–PrxVI), which are usually present at high intracellular concentrations and have been implicated in a variety of cellular processes, including cell proliferation [47]. The structure of subcellular organelles is changed dynamically throughout the cell cycle, and therefore, the localizations of Prxs are reorganized depending on cell cycle phases.

Mammalian Prx isoforms localize to specific cellular compartments, including the cytosol, and to organelles such as mitochondria, peroxisomes, lysosomes, and the nucleus, as well as localizing outside the cell [13,47,48,49] (Figure 2, Table 1). PrxI was localized in moderate amounts in the nucleus, cytosol, and plasma membrane [50,51]. In addition, a recent study described local regulation of H_2_O_2_ around the centrosome through PrxI phosphorylation by Cdk1 during early mitosis [17]. PrxII was observed primarily in the cytosol, with distribution similar to PrxI, although some was found in the nucleus and involved in protection of cancer cell death from DNA damage [52]. PrxIII was found almost exclusively in the matrix of mitochondria, which is mainly responsible for reversible regulation of mitochondrial H_2_O_2_ levels [53,54]. While PrxIV was found in both cytosol and extracellular fluid, it was the only isoform found in the ER [55,56]. Protein disulfide isomerase was oxidized by ER-localized PrxIV in the oxidative protein-folding pathway [57,58,59]. PrxV was found predominantly in the mitochondria, with some in the peroxisome, and low levels in cytosol and nucleus [60,61,62]. PrxV antioxidant enzyme protects against oxidant-generating peroxisome, containing several molecules that catalyze the oxidation of substrates and thereby produce H_2_O_2_ [63]. PrxVI was located predominantly in the cytosol, but it was also expressed in lysosomal compartments with PrxIV [49,64]. Phosphorylation of PrxVI by the MAP kinase increases its PLA_2_ (phospholipase A_2_) activity [65,66,67].

## 4. Oscillation of the Intracellular Concentration of H_2_O_2_ during the Cell Cycle

ROS levels were shown previously to change during cell cycle progression when measured with the fluorescent probe chloromethyl derivative of 2′,7′-dichlorodihydrofluorescein diacetate [9,15]. This probe reacts with various oxidants including H_2_O_2_, the hydroxyl radical, NO, lipid peroxides, and peroxynitrite [9]. Furthermore, given that the oxidation reaction of the probe is irreversible, the change in fluorescence represents the sum of the oxidant molecules produced after introduction of the probe—not transient changes in oxidant concentration. We measured intracellular H_2_O_2_ levels in HeLa cells at various cell cycle stages with the use of a genetically encoded fluorescent probe, HyPer (Evrogen, Russia), that reacts reversibly and specifically with H_2_O_2_ [68]. The relative fluorescence level in nocodazole-arrested prometaphase cells (~98% in the G_2_–M phase) was about three times that in asynchronous cells (~73% in G_1_, ~14% in S, and ~13% in G_2_–M), whereas the level in G_1_-enriched cells (~52% in G_1_, ~1% in S, and ~47% in G_2_–M) was similar to that in asynchronous cells (Figure 3A,B). The fluorescence of HyPer is sensitive to changes in pH, and a HyPer mutant with Cys^199^ replaced by Ser (HyPer–C199S) is sensitive to pH but not to H_2_O_2_. The fluorescence of HyPer–C199S in HeLa cells was not affected by nocodazole arrest (Figure 3C). Release of cells from G_1_–S arrest also revealed that HyPer fluorescence intensity increased as cells progressed through S phase, reached a maximum when the G_2_–M population was largest, and then decreased as cells transited into G_1_ phase (Figure 3D–F). Linear regression analysis of the fluorescence intensities measured at six different time points after the release of cells from G_1_–S arrest yielded a ratio of 1:1.7:2.9 for the relative H_2_O_2_ levels in G_1_, S, and G_2_–M phases, respectively (adjusted *R*^2^ value = 0.982, *p* = 0.00145). These results indicate that endogenous H_2_O_2_ levels oscillate in a cell cycle-dependent manner. Oscillations in thiols and their reductive capacity could lead to increased HyPer oxidation.

## 5. Source for H_2_O_2_ Generation throughout the Cell Cycle

Energy metabolism, which is accompanied by H_2_O_2_ production, increases during cell proliferation; H_2_O_2_ derived from NADPH oxidase has been shown to be required for G_2_–M progression [10]; cytosolic phospholipase A_2_ activity, which generates arachidonic acid, is high in mitotic phase and decreases in G_1_ phase [69]; and arachidonic acid-metabolizing enzymes such as cyclooxygenase, lipoxygenase, and cytochrome P450 generate H_2_O_2_ [70]. There are thus multiple potential sources of H_2_O_2_ production during cell cycle progression. The general flavoprotein inhibitor diphenyleneiodonium, the cytosolic phospholipase A_2_ inhibitors arachidonyl trifluoromethyl ketone and methyl arachidonyl fluorophosphonate, and the lipoxygenase inhibitor nordihydroguaiaretic acid each inhibited entry into mitosis in HeLa cells [17]. This result suggests that NADPH oxidases and the cytosolic phospholipase A_2_–lipoxygenase system are responsible for the production of H_2_O_2_ at G_2_–M transition.

Specific Cdks control structural change in cellular organelles such as the Golgi, ER, and mitochondria and thereby allow segregation between daughter cells of each organelle during mitosis [34,45]. Golgi H_2_O_2_ is shown to be produced via Ca^2+^-dependent Duox in epidermal growth factor-stimulated cells [35]. Some portion of fragmented Golgi is redistributed into the ER containing a high concentration of Ca^2+^ [71] and the production of H_2_O_2_ via Duox is predicted at mitotic phase. Mild or severe oxidative stress induces mitochondrial fission and fragmentation [72]. Fragmented mitochondria can be degraded selectively by mitophagy to protect the cell from apoptosis. Granular (fragmented) mitochondria are linked to high levels in ROS, and tubular (filamentous) mitochondria are inversely proportional to ROS level in neurodegeneration [73]. During mitosis, the question of whether fragmented mitochondria produce H_2_O_2_ more than filamentous mitochondria should be addressed.

## 6. Protective and Signaling Role of Peroxiredoxin around the Centrosomes

Local inactivation of PrxI is important for the transduction of H_2_O_2_ signal in addition to local generation of H_2_O_2_ by an activated NADPH oxidase during extracellular stimulation. PrxI is phosphorylated on Tyr^194^ by a Src family kinase, and therefore, the inhibition of its peroxidase activity allows a transient increase in H_2_O_2_ level in the plasma membrane microdomain in cells stimulated with growth factors such as platelet-derived growth factor and epidermal growth factor [50].

Centrosomes are microtubule-organizing centers that play key roles in spindle formation, chromosome segregation, cell cycle progression, and cell division throughout the cell cycle [74]. In a recent study, we proposed a role for PrxI associated with centrosomes for mitotic entry [17] (Figure 4). Many proteins that participate in the mitotic entry network (Cdk1, cyclin B, Cdc25B, Cdc25C, Plk1, and Aurora A) or in the mitotic exit network (APC/C, Cdc14B, and Cdh1) are concentrated at the centrosome [19,75,76,77,78,79,80]. Several feedback loops that underlie activation of Cdk1 and amplification of Cdk1 activity are initiated by cyclin B, Cdc25B, Cdc25C, Plk1, and Aurora A at the centrosome. Mitotic exit processes related to the activation of APC/C–Cdh1 by Cdc14B, which results in the degradation of mitotic activators (cyclin B, Plk1, and Aurora A), also have been suggested to be integrated at the centrosome [81].

The activity of Cdc14B must be controlled tightly to avoid premature degradation of mitotic activators before full activation of Cdk1. Our results indicate that the activity of centrosomal Cdc14B is controlled through regulation of the local H_2_O_2_ concentration around the centrosome, which in turn depends on the H_2_O_2_-eliminating enzyme PrxI. PrxI is present at the centrosome, and the centrosome-associated PrxI, but not cytosolic PrxI, is specifically phosphorylated on Thr^90^, a consensus site for phosphorylation by Cdk1 [17,82]. Phospho-PrxI was detected at the centrosome of HeLa cells during early mitosis (prophase, prometaphase, and metaphase) but not during interphase (G_2_) or late mitotic stages (anaphase, telophase, and cytokinesis). Phosphorylation of PrxI inactivates its peroxidase function, resulting in exposure of centrosomal Cdc14B to H_2_O_2_-dependent inactivation. This inactivation of Cdc14B likely blocks untimely dephosphorylation of Cdh1 during early mitosis and thereby prevents premature degradation of the Cdk1 activators cyclin B, Plk1, and Aurora A by APC/C–Cdh1. Whether centrosomal proteins are oxidized is difficult to investigate because the biochemical procedure to isolate the centrosome includes treatment with a thiol-reducing agent, such as 2-mercaptoethanol [80], which reduces the disulfide bond of oxidized proteins. Instead, the specific role of pericentrosomal H_2_O_2_ was demonstrated by targeted expression of catalase at the centrosome, which resulted in downregulation of the centrosomal abundance of cyclin B1, Plk1, and Aurora A by ~30%–55% and delayed mitotic entry. Targeted expression of catalase also reduced the cellular level of Ser^40^-phosphorylated Cdh1 by ~30%. The Ser^40^-phosphorylated form of Cdh1 was not detected reliably in isolated centrosomes, probably because phosphorylation reduces the affinity of Cdh1 for the organelle to a level insufficient to support the interaction during biochemical isolation. These results, together with the high sensitivity of Cdc14B to H_2_O_2_, thus support the notion that changes in pericentrosomal H_2_O_2_ level associated with cell cycle progression markedly influence the stability of cyclin B1, Plk1, and Aurora A through the Cdc14B and Cdh1–APC/C axes.

Our proposal for sequential events determined by inactivation of PrxI via Cdk1 at the centrosome is shown in a schematic drawing (Figure 5). Intracellular H_2_O_2_ levels peak when the G_2_–M population is largest (see Figure 3D,E). In the G_2_ phase, PrxI associated with the centrosome shields the organelles from the high tide of H_2_O_2_. As the abundance of cyclin B peaks in late G_2_ phase and early mitosis, cyclin B binds to and activates Cdk1. The activity of Cdk1–cyclin B is then further increased through the operation of multiple feedback loops mediated by cyclin B, Cdc25, Plk1, and Aurora A. At the same time, Cdk1–cyclin B phosphorylates (inactivates) PrxI, which results in an increase in H_2_O_2_ level in the centrosome and consequent inactivation of Cdc14B. In the absence of the H_2_O_2_-mediated inactivation of Cdc14B, which is highly enriched at the centrosome, Cdc14B is expected to dephosphorylate Cdk1-phosphorylated Cdh1 and to thereby activate APC/C–Cdh1. These events in turn are expected to result in degradation of the mitotic activators (cyclin B, Cdc25, Plk1, and Aurora A) and to retard mitotic entry. Given that Cdc25 is not oxidized by H_2_O_2_, the Cdk1 activation loop mediated by Cdc25B and Cdc25C is not affected by PrxI inactivation. In late mitosis, PrxI is dephosphorylated and H_2_O_2_ is removed by active PrxI at the centrosome, which induces sequential reactivation of Cdc14B and activation of APC/C–Cdh1 by means of dephosphorylation of Cdh1.

Protein phosphatase 1 (PP1) or PP2A (or both) is likely responsible for the dephosphorylation of PrxI during late mitosis, given that we observed that phosphorylated PrxI accumulates in HeLa cells treated with okadaic acid, a specific inhibitor of these phosphatases [82]. This conclusion is consistent with previous observations that PP1 and PP2A are key mitotic exit phosphatases that counteract phosphorylation by Cdk1 [83]; that PP1 and PP2A, together with their regulatory proteins, are concentrated at the centrosome [84,85,86,87]; and that okadaic acid promotes mitotic entry through inhibition of these phosphatases [88].

While APC/C–Cdh1 remains active in G_1_ and G_0_ (quiescent phase) cells, Cdks must be inactive. Here, APC/C–Cdh1 blocks the premature accumulation of certain positive regulators of S-phase and mitotic progression. Since H_2_O_2_ can trigger S-phase entry and mitotic signaling by inactivating Cdc14B and APC/C–Cdh1 at the centrosome, centrosome-associated PrxI can prevent unscheduled mitotic entry as a result of any accidental increase in H_2_O_2_ (such as in response to ultraviolet or ionizing radiation) (Figure 4). This corresponds with previous reports that Cdc14B, Cdh1, and PrxI function as tumor suppressors [80,89,90].

In brief summary (Figure 4 and Figure 5), Cdc14B is coupled to and inactivated by a high tide of cytosolic H_2_O_2_ through Cdk1-dependent phosphorylation (inactivation) of PrxI at the centrosome during the early stage of mitosis, resulting in the formation of a previously unknown feedback loop in which Cdk1 activates itself by blocking the degradation of its positive regulators. During mitotic exit, however, Cdc14B is decoupled from cytosolic H_2_O_2_ as a result of PrxI dephosphorylation and is reactivated to promote the degradation of the mitotic activators. The new finding reveals that H2O2 molecules from diverse cellular processes, including mitochondrial respiration, arachidonic acid metabolism, and activated NADPH oxidase in fragmented organelles (the Golgi or ER), control mitosis.

In the model (Figure 4 and Figure 5), given that the peroxidase activity of phosphorylated PrxI by Cdk1 is about 20% of that of unphosphorylated PrxI [82], a local increase in H_2_O_2_ levels at the centrosomes is reasonably expected to oxidize redox-sensitive proteins represented by Cdc14B. PrxI acts as an H_2_O_2_ floodgate, preventing susceptible targets from oxidation by H_2_O_2_ until the floodgate is opened [91]. However, if the residual activity of p-PrxI is enough to remove H_2_O_2_ at the centrosomes or if other cytosolic Prxs replace inactivated PrxI, then Prxs-dependent redox relay for mitotic progression can be considered.

## 7. Possible Role of Peroxiredoxin around the Golgi, ER, and Mitochondria

The intracellular concentration of H_2_O_2_ oscillates during cell cycle progression, and the sources of this H_2_O_2_ include NADPH oxidase and arachidonic acid-metabolizing enzymes [17]. The best characterized H_2_O_2_ effectors are members of the protein tyrosine phosphatase (PTP) family such as PTP1B and phosphatase and tensin homologue (PTEN) [2,92,93]. Indeed, in cells stimulated with various growth factors, the activation of protein tyrosine kinases or phosphatidylinositol 3-kinase is not sufficient to increase the steady-state level of tyrosine-phosphorylated proteins or 3-phosphorylated inositides; the concurrent inhibition of PTPs or of PTEN, respectively, is also required. Given that many kinases (such as Src family members) and transcription factors (such as p53 and AP1) are also direct targets of H_2_O_2_ [2,92,94], oscillation of the intracellular H_2_O_2_ concentration is expected to affect cell cycle progression at multiple stages.

Considering a dynamic change in cell morphology, including breakdown of the nuclear envelope and fragmentation of subcellular organelles such as the Golgi, ER, and mitochondria in mitosis, Prx isoforms are redistributed in the cell throughout the cell cycle. PrxI associated with the ER membrane functions in attenuating oxidative stress in the livers of ethanol-fed mice [95]. The cell at the G_2_–M transition and in mitotic phase requires an abundance of energy to drive dynamic change in cell morphology under oxidative stress (with high H_2_O_2_ levels). To reduce the requirement of energy, mitotic cells likely perform H_2_O_2_-mediated signaling more efficiently than interphase cells, which needs to be verified in further investigations. Cytosolic Prxs, subcellular organelle-associated Prxs, and Prxs in the lumen of organelles can play key roles in controlling H_2_O_2_-mediated signaling, as in H_2_O_2_ accumulation by the phosphorylation (inactivation) on Tyr^194^ of PrxI at the plasma membrane [50] and by the phosphorylation (inactivation) on Thr^90^ of PrxI at the centrosomes [17].

## 8. Conclusions

Abundant Prx proteins react with H_2_O_2_ molecules and thereby act as sensors and transducers of H_2_O_2_-mediated signaling. The localization of Prxs is changed dynamically depending on structural reorganization of subcellular organelles throughout the cell cycle. The level of ROS, especially H_2_O_2_ molecules, fluctuates throughout the cell cycle, is lowest in G_1_ phase, and peaks in mitosis as shown in this review or other previous studies [9,15]. Given that Prxs protect cells from oxidative stress by removing H_2_O_2_ molecules, Prxs ensure proper cell cycle progression in a time-dependent manner by inhibiting premature activation of Cdk1, as proposed for PrxI at the centrosome [17]. Prxs protect the cell from premature cell cycle progression under unwanted oxidative stress arising by UV or ionizing irradiation in interphase. Downregulation and overexpression of PrxII or PrxIII elicits a change in the distribution of cell cycle stages [8,96,97], indicating Prxs are critical for proper transition of each stage. Modulation of the expression of Prx isoforms has limitations for investigation of its role as the regulator of cell cycle. Identification of H_2_O_2_ effectors in mitotic cells with high levels of H_2_O_2_ and oxidatively damaged cells that are derived from cardiovascular diseases and neurodegenerative disorders sheds light on understanding the roles of Prxs during cell cycle progression.

Proper and ordered cell progression requires the accurate regulation of Cdks and APC/C activities. Among the regulators of Cdks–APC/C module, Cdc14B is known to be an H_2_O_2_ effector [17]. The Ser/Thr phosphatases, PP1 and PP2, which play a critical role in mitotic entry and exit in mammalian cells [88,98,99,100,101,102], are good candidates for H_2_O_2_ effectors because these proteins have a redox-sensitive catalytic center [103,104,105]. The phosphorylation (inactivation) of PrxI by Cdk1 at the centrosomes induces APC/C inactivation through releasing an activator, Cdh1 (Figure 5) [17]. Another example of the role of Prx in the regulation of Cdks-APC/C module is a phosphorylation (inactivation) on Thr^89^ of PrxII by Cdk5 in Parkinson’s disease (PD) [106]. Oxidative stress is a critical factor to induce cellular damage in PD [107]. Increased H_2_O_2_ by Cdk5-mediated PrxII inactivation in PD occurs with APC/C-Cdh1 inhibition by Cdk5 [108]. It is suggested that the Cdk–Prx–APC/C axis is conserved in different cellular events.

## Figures and Tables

**Figure 1 antioxidants-09-00280-f001:**
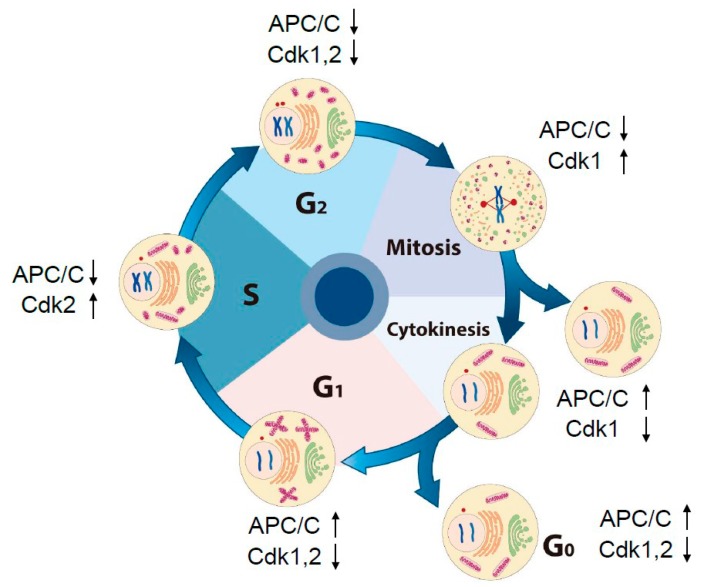
The morphological change in subcellular compartments and the activities of cell cycle-dependent kinases (Cdks) and anaphase-promoting complex/cyclosome (APC/C) throughout the cell cycle in a mammalian cell: During the cell cycle, after DNA replication is finished at the S phase, a cell prepares to enter mitosis through the G_2_ phase. At the G_2_–M transition, the structure of subcellular organelles is changed dramatically as illustrated here. A nuclear envelop is broken and the Golgi (green), ER (yellow), and mitochondria (pink) are fragmented. After mitotic exit, all the organelles are reorganized and structured in filamentous states. Activity of APC/C is inversely propositioned to that of Cdks during cell cycle progression except at the G_2_ phase. APC/C activity is inhibited by early mitotic inhibitor1 at the G_2_ phase. In G_0_ (quiescent phase) cells, APC/C–Cdh1 remains active.

**Figure 2 antioxidants-09-00280-f002:**
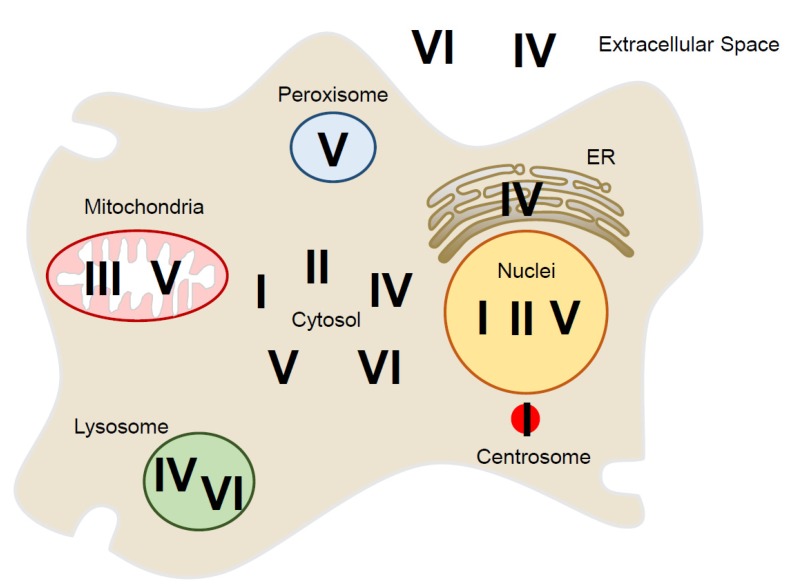
Localization of six peroxiredoxin isoforms (PrxI–PrxVI) inside and outside of the mammalian cell: In the cytosolic space, PrxI, II, IV, V, and VI are present. The nucleus has PrxI, PrxII, and PrxV, and mitochondria have PrxIII and PrxV. In the peroxisome, PrxV exists, and in the ER, PrxIV is present. The lysosome has PrxIV and PrxVI. PrxI is also associated with the centrosome. PrxIV and PrxVI are present outside of the cell.

**Figure 3 antioxidants-09-00280-f003:**
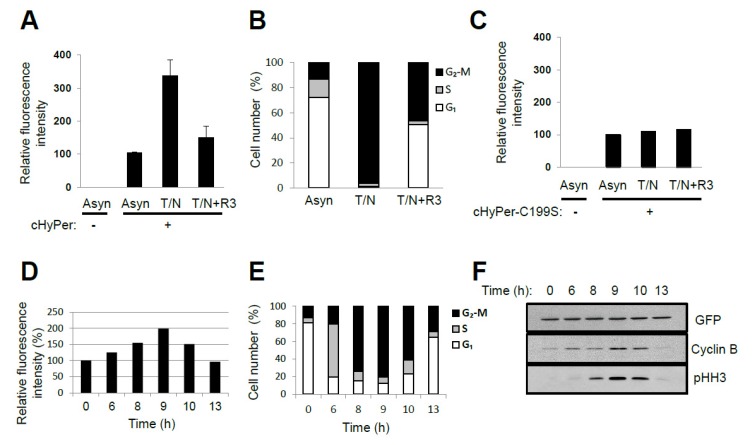
Oscillation of H_2_O_2_ levels during the cell cycle (experimental data by the authors): (**A**,**B**) HeLa cells expressing cytosolic HyPer (cHyPer) were arrested at prometaphase by treatment with thymidine and nocodazole (T/N) and then released into G_1_ phase by shaking off into fresh medium and incubation for 3 h (T/N+R3). The fluorescence intensity of cHyPer was measured by flow cytometry, and the relative levels of H_2_O_2_ were estimated (**A**) for asynchronous, T/N, and T/N+R3 cells. Asynchronous HeLa cells that were not transfected with the cHyPer vector were analyzed similarly. Cell cycle status was also determined by flow cytometric analysis after staining with propidium iodide (**B**). Data in (**A**) are means ± SD from three independent experiments. (**C**) The relative fluorescence intensity of cHyPer–C199S was measured as in (**A**). Data are from a representative experiment. (**D–F**) HeLa cells expressing cHyPer were arrested at G_1_–S with a double-thymidine block and released into fresh medium. The relative levels of H_2_O_2_ were measured by flow cytometry at the indicated times after the release (**D**). Cell cycle status was also analyzed by flow cytometry after staining with propidium iodide (**E**), and the expression of cell cycle marker proteins was examined by immunoblot analysis (**F**). cHyPer is detected with antibodies to green fluorescent protein. Data are from a representative experiment. pHH3, phosphorylated histone H3. Rabbit polyclonal antibodies to Ser^10^-phosphorylated histone H3 (pHH3) (from Upstate Biotechnology) were used to detect mitotic cells. For analysis of cell cycle stage, cells (5 × 10^5^/mL) were washed twice with ice-cold phosphate-buffered saline, fixed overnight at 4 °C in 70% ethanol, and stained with 1 mL of a solution containing RNase (50 μg/mL) and propidium iodide (50 μg/mL) before flow cytometry with a FACSCalibur instrument (BD Biosciences, Franklin Lakes, USA). For measurement of the intracellular level of H_2_O_2_, cells expressing cHyPer or cHyPer-C199S were analyzed at an excitation wavelength of 488 nm.

**Figure 4 antioxidants-09-00280-f004:**
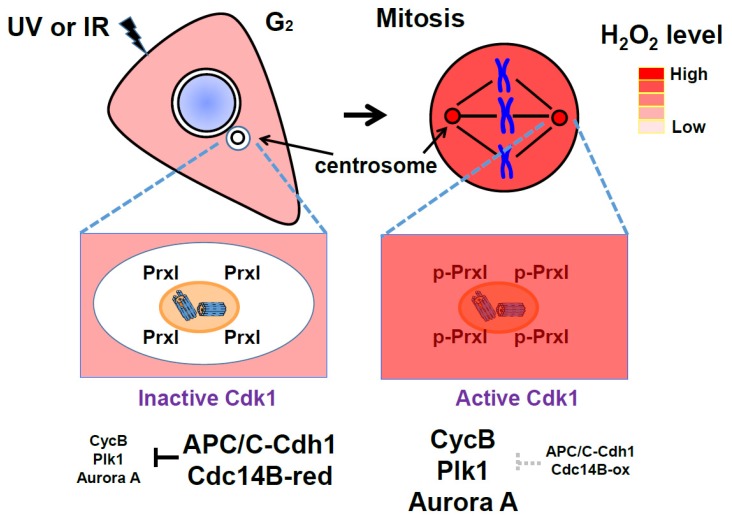
Protective role of peroxiredoxin I (PrxI) in the cell under oxidative stress: Centrosome-associated PrxI proteins shield the organelles from a high tide of H_2_O_2_ molecules from various intracellular sources at G_2_ phase. Anaphase-promoting complex/cyclosome (APC/C)–Cdh1 activity is high at the centrosome, and therefore, its substrates such as cyclin B, polo-like kinase1 (Plk1), and Aurora A are degraded. During mitosis, PrxI is inactivated by cell cycle-dependent kinase 1 (Cdk1) through phosphorylation and the large numbers of H_2_O_2_ molecules inhibit APC/C-Cdh1, which results in increased cyclin B, Plk1, and Aurora A proteins. Centrosome-associated PrxI can protect cells from other oxidative stress, including ultraviolet (UV) or ionizing radiation (IR).

**Figure 5 antioxidants-09-00280-f005:**
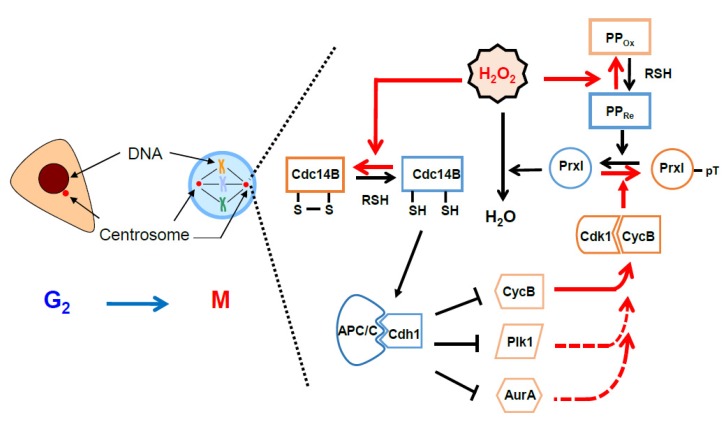
A model illustrating the role of pericentrosomal H_2_O_2_ for activation of cell cycle-dependent kinase 1 (Cdk1): Red arrows indicate the direction of the positive feedback reactions resulting from peroxiredoxin I (PrxI) phosphorylation by Cdk1 at the mitotic centrosomes. Black arrows indicate the direction of those resulting from PrxI dephosphorylation. Dashed red arrows indicate multiple positive feedback loops known to activate Cdk1 directly or indirectly. AurA, Aurora A; CycB, cyclin B; Plk1, polo-like kinase1; APC/C, anaphase-promoting complex/cyclosome; PP, protein phosphatase; Cdc14B, cell division cycle 14B. See Section 6 for details.

**Table 1 antioxidants-09-00280-t001:** Subcellular localization of peroxiredoxin isoforms inside and outside of the cell.

Prx Isoform	Subfamily	Subcellular Localization	Reference
PrxⅠ	2-Cys Prx	Cytosol, nucleus, centrosome, plasma membrane	[13,17,47,48,49,50,51]
PrxⅡ	2-Cys Prx	Cytosol, nucleus, plasma membrane	[13,47,48,49,52]
PrxⅢ	2-Cys Prx	Mitochondria	[13,47,48,49,53,54]
PrxⅣ	2-Cys Prx	Cytosol, ER, lysosome, extracellular localization	[13,47,48,49,55,56,57,58,59]
PrxⅤ	Atypical 2-Cys Prx	Cytosol, mitochondria, peroxisome, nucleus	[13,47,48,49,53,60,61,62,63]
PrxⅥ	1-Cys Prx	Cytosol, lysosome, extracellular localization	[13,47,48,49,64,65,66,67]

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
