# Peer review of "The Role of Hydrogen Peroxide and Peroxiredoxins throughout the Cell Cycle"

_antioxidants, 2020, doi:10.3390/antiox9040280_

Round 1

Reviewer 1 Report

This is a well written review, with my comments relatively minor in nature. My main general comment is that the authors fail to portray some of the uncertainty associated with redox signalling, and I challenge them to describe some of the limitations of the work they describe (see specific comments below).

1. I recommend the title is changed to “The role of hydrogen peroxide and peroxiredoxins throughout the cell cycle”. Changes in hydrogen peroxide levels and the consequences of this feature prominently throughout the review, and are of interest in their own right.

2. Line 11: change “H2O2 effectors” to “thiol proteins”

3. Line 14: should be spelt “peroxynitrite”.

4. Line 14-16: this floodgate-model is only one mechanism by which peroxiredoxins are proposed to influence signalling. Redox relays are another, and deserve a mention.

5. Line 18: the review also summarises changes in hydrogen peroxide levels and the consequences of this (see recommended change to title).

6. Line 20-22: change “molecules” to “levels” and stop the sentence there. Speculation on sources is best left to the main text of the review.

7. Line 27: delete “ageing” as it isn’t well established that H2O2 plays a signalling role in ageing.

8. Line 32: needs an explanation of how extracellular RNS result in H2O2 production.

9. Line 40: change to “abundant intracellular”

10. Line 43: the Winterbourn group has shown that glutaredoxin reduces peroxiredoxins, not glutathione.

11. Line 82-84: the limitations of the DCF assay are well known, and can change due to a number of factors other than increased H2O2. Discuss.

12. Line 177-178: in addition to H2O2, oscillations in thiols and reductive capacity could lead to increased HyPer oxidation.

13. Section 5: inhibitors are not selective, especially DPI which targets flavoproteins in general, not just NOX’s. Mention this uncertainty.

14. Section 6: mention that phosphorylation lowers the reactivity of peroxiredoxins, but does not eliminate it, so interpretation is not as simple as portrayed. Discuss.

15. Section 6: where is Prx2 located, and does this assist in reducing H2O2 levels when Prx1 is inhibited? Also, there are modelling studies that indicate the rapid diffusion of H2O2 and consumption by enzymes elsewhere in the cell means that local levels do not build significantly. This is a limitation of the floodgate model that deserves mention.

16. Lines 267-268: peroxiredoxins do not have a “high-affinity binding site”, but structural features enhance reactivity with H2O2. Also, catalase is effective at low concentrations of H2O2 – the difference with peroxiredoxins is that catalase maintains reactivity at high concentrations that inactivate peroxiredoxins. I don't think this argument can explain the lack of effect of catalase. A peroxiredoxin-dependent redox relay could be considered.

Author Response

This is a well written review, with my comments relatively minor in nature. My main general comment is that the authors fail to portray some of the uncertainty associated with redox signalling, and I challenge them to describe some of the limitations of the work they describe (see specific comments below).

  1. I recommend the title is changed to “The role of hydrogen peroxide and peroxiredoxins throughout the cell cycle”. Changes in hydrogen peroxide levels and the consequences of this feature prominently throughout the review, and are of interest in their own right.

We changed the title as suggested by reviewer #1 (Line 2).

  1. Line 11: change “H2O2 effectors” to “thiol proteins”

We changed it (Line 12).

  1. Line 14: should be spelt “peroxynitrite”.

We changed it (Line 15).

  1. Line 14-16: this floodgate-model is only one mechanism by which peroxiredoxins are proposed to influence signalling. Redox relays are another, and deserve a mention.

We added the description “Given that Prxs are highly sensitive to oxidation by H2O2, they act as sensors and transducers of H2O2 signaling via transferring their oxidation state to effector proteins” (Line 18-19).

  1. Line 18: the review also summarises changes in hydrogen peroxide levels and the consequences of this (see recommended change to title).

We changed the title and added the consequence of changes of H2O2 levels as suggested by reviewer #1 (Line 2, 22).

  1. Line 20-22: change “molecules” to “levels” and stop the sentence there. Speculation on sources is best left to the main text of the review.

We changed it (Line 24).

  1. Line 27: delete “ageing” as it isn’t well established that H2O2 plays a signalling role in ageing.

We deleted it (Line 29).

  1. Line 32: needs an explanation of how extracellular RNS result in H2O2 production.

We added the explanation (Line 34-35).

  1. Line 40: change to “abundant intracellular”

We changed it (Line 43).

  1. Line 43: the Winterbourn group has shown that glutaredoxin reduces peroxiredoxins, not glutathione.

We changed it and added a reference (Line 46).

  1. Line 82-84: the limitations of the DCF assay are well known, and can change due to a number of factors other than increased H2O2. Discuss.

We added the limitations of DCF probe (Line 87-89) and also discussed them more in section 4 (Line 167-170).

  1. Line 177-178: in addition to H2O2, oscillations in thiols and reductive capacity could lead to increased HyPer oxidation.

We added the sentence (Line 185-186).

  1. Section 5: inhibitors are not selective, especially DPI which targets flavoproteins in general, not just NOX’s. Mention this uncertainty.

We added the uncertainty of DPI usage (Line 215-216, 218).

  1. Section 6: mention that phosphorylation lowers the reactivity of peroxiredoxins, but does not eliminate it, so interpretation is not as simple as portrayed. Discuss.

We added the discussion in Section 6 (Line 328-334).

  1. Section 6: where is Prx2 located, and does this assist in reducing H2O2 levels when Prx1 is inhibited? Also, there are modelling studies that indicate the rapid diffusion of H2O2 and consumption by enzymes elsewhere in the cell means that local levels do not build significantly. This is a limitation of the floodgate model that deserves mention.

Among the four cytosolic Prx enzymes (PrxI, II, V, and VI), only PrxI was detected at the centrosome [Lim, J.M et.al. J Cell Biol 2015, 210, 23-33]. We added the discussion in Section 6 (Line 328-334).

  1. Lines 267-268: peroxiredoxins do not have a “high-affinity binding site”, but structural features enhance reactivity with H2O2. Also, catalase is effective at low concentrations of H2O2 – the difference with peroxiredoxins is that catalase maintains reactivity at high concentrations that inactivate peroxiredoxins. I don't think this argument can explain the lack of effect of catalase. A peroxiredoxin-dependent redox relay could be considered.

We removed the following sentence “Catalase is not an ideal substitute for PrxI because catalase cannot remove low levels of H2O2 as efficiently as PrxI—which has a high-affinity binding site for H2O2 [39]—possibly explaining why the inhibition of mitotic function by expression of centrosome-targeting catalase was not more extensive.” (Line 267-270) and added the discussion in Section 6 (Line 328-334).

Reviewer 2 Report

This manuscript by Heo et al. summarizes information about H2O2 levels and peroxiredoxin involvement with cell cycle, including some previously unreported data that uses Hyper and synchronized cells to better address the question of H2O2 levels.  Overall I find this to be a very useful review.  I wonder whether inclusion of new data is allowed, and I do recommend including a little more information about the experimental work since there is no methods section.  Otherwise, I think it is a good contribution.

Major point

If the new data of Fig. 3 is allowed, it would be helpful to say a bit more about how the experiments were done.  What is definitely needed is an indication of what is meant by “pHHE” in panel F.

I am confused by the statement in lines 282/283 that Cdc25 is not oxidized by H2O2, which seems to be contradicted by the statements in lines 358/359 that they are known to be H2O2 effectors.

Minor points

Line 145-146, this should be reworded to not imply that cancer cells purposefully puts PrxI into “the nucleus to protect cancer cell death from DNA damage”.  The next sentence has an awkward and I think incorrect phrase of “reversible metabolizing”.

Line 161, I suggest adding “of the probe” after “the oxidation reaction” to enhance clarity.

Lines 201/202 and 204, DPI is not precisely a “NADPH oxidase inhibitor” but rather a general flavoprotein inhibitor.  So in the following sentence the word “indicates” is too strong and, even given “selective” inhibitors, should be replaced with “suggests”.

Lines 214 and 215, the word “mitochondria” is plural.

I would suggest including reference #89 (Phalen et al.) with citation #4 in the introduction.

“Peroxynitrite” is misspelled in the abstract.  “Try194” of line 338 should by Tyr194.

Author Response

This manuscript by Heo et al. summarizes information about H2O2 levels and peroxiredoxin involvement with cell cycle, including some previously unreported data that uses Hyper and synchronized cells to better address the question of H2O2 levels.  Overall I find this to be a very useful review.  I wonder whether inclusion of new data is allowed, and I do recommend including a little more information about the experimental work since there is no methods section.  Otherwise, I think it is a good contribution.

Major point

If the new data of Fig. 3 is allowed, it would be helpful to say a bit more about how the experiments were done.  What is definitely needed is an indication of what is meant by “pHHE” in panel F.

We added a definition for pHH3 and a brief experimental procedure in the legend of Figure 3 (Line 202-208).

I am confused by the statement in lines 282/283 that Cdc25 is not oxidized by H2O2, which seems to be contradicted by the statements in lines 358/359 that they are known to be H2O2 effectors.

We changed the statement in Section 8 Conclusion (Line 374).

Minor points

Line 145-146, this should be reworded to not imply that cancer cells purposefully puts PrxI into “the nucleus to protect cancer cell death from DNA damage”.  The next sentence has an awkward and I think incorrect phrase of “reversible metabolizing”.

We changed the sentences (Line 151-154).

Line 161, I suggest adding “of the probe” after “the oxidation reaction” to enhance clarity.

I added it (Line 168).

Lines 201/202 and 204, DPI is not precisely a “NADPH oxidase inhibitor” but rather a general flavoprotein inhibitor.  So in the following sentence the word “indicates” is too strong and, even given “selective” inhibitors, should be replaced with “suggests”.

We replaced it with “general flavoprotein inhibitor” (Line 215-216) and replaced “indicates” with “suggests” (Line 218).

Lines 214 and 215, the word “mitochondria” is plural.

We changed with “mitochondria are” (Line 228).

I would suggest including reference #89 (Phalen et al.) with citation #4 in the introduction.

We added the reference (Phalen et al.) (Line 37).

“Peroxynitrite” is misspelled in the abstract.  “Try194” of line 338 should by Tyr194.

We changed them (Line 15, 355).

We believe that our point-by-point responses address all concerns raised by the reviewers. We make minor changes in our revised manuscript as suggested by the reviewers.